Drivers and decoupling analysis of carbon emissions in the non-ferrous metal industry-evidence from 28 provinces in China

Zeng Guohua 1
Zhong Minglong 919505253@qq.com 1
Xiao Chengzhang 2
1 School of Economics and Management, JiangXi University of Science and Technology , Ganzhou , Jiangxi , China
2 Monash University , Melbourne , Australia
Mahmood Haider
Electronic publication date: 2023 Dec 14
Publication date: 2023
Volume: 11
Electronic Location ID: e16575
Received 2023 Jul 27; Accepted 2023 Nov 13
Copyright: ©2023 Zeng et al.
Copyright year: 2023
Copyright holder: Zeng et al.
License: This is an open access article distributed under the terms of the Creative Commons Attribution License, which permits unrestricted use, distribution, reproduction and adaptation in any medium and for any purpose provided that it is properly attributed. For attribution, the original author(s), title, publication source (PeerJ) and either DOI or URL of the article must be cited.
License URL: https://creativecommons.org/licenses/by/4.0/

Keywords: Non-ferrous metals industry, Carbon emissions, Production, Decoupling, Tapio model

Funding: National Social Science Foundation of China 21XGL02 18BGL246 Youth Jinggang Scholars Program in Jiangxi Province QNJG2020047 Jiangxi Higher Education Teaching Reform Research Project JXJG-20-7-32 Innovation and Entrepreneurship Training Program for College Students 202010407029 This study was financially supported by the National Social Science Foundation of China (No: 21XGL02, 18BGL246), the Youth Jinggang Scholars Program in Jiangxi Province (No: QNJG2020047), the Jiangxi Higher Education Teaching Reform Research Project (No: JXJG-20-7-32), and the Innovation and Entrepreneurship Training Program for College Students (No: 202010407029). The funders had no role in study design, data collection and analysis, decision to publish, or preparation of the manuscript.

==============================
Emissions from the non-ferrous metal industry are a major source of carbon emissions in China. Understanding the decoupling of carbon emissions from the non-ferrous metal industry and its influencing factors is crucial for China to achieve its “double carbon” goal. Here, we applied the Tapio decoupling model to measure the decoupling status and developmental trends of carbon output and emissions of the non-ferrous metal industry in China. The panel interaction fixed effects model is used to empirically analyze the influencing factors of carbon emissions in China’s non-ferrous metal industry. The results show that carbon emissions from China’s non-ferrous metal industry have experienced three main states: strong decoupling, growth connection, and negative growth decoupling. The carbon emissions of the non-ferrous metal industry in some eastern and central provinces from 2000 to 2004 were in a negative decoupling state. Most provinces in the western and central regions were either in a strong or weak decoupling state based on the developmental trend of the decoupling state of carbon emissions. However, from 2015 to 2019, the decoupling status of carbon emissions in most provinces in western and central China had a significantly negative, weakly negative, or a negative growth decoupling status. Energy structure, energy intensity, cost, and non-ferrous metal production all have a positive driving effect on carbon emissions in the non-ferrous metal industry. Production had a mitigating effect on carbon emissions in the non-ferrous metal industry between 2010–2014 in the eastern region of China. From the results of our study, we propose policy recommendations to promote a strong decoupling of carbon emissions from the non-ferrous metal industry by improving energy structure, reducing energy intensity, and optimizing production capacity.

Introduction

China has become the world’s largest consumer of non-ferrous metals. In 2020, the output of China’s non-ferrous metals industry reached 61.68 million tons, which was an increase of 5.5 percent over the previous year. As an industry with high energy-consumption and high carbon-emissions, its related CO2 emissions have risen due to its increasing scale and the predominance of thermal power as a source of electricity. The total energy consumption for this industry is approximately 250 million tons of standard coal with a carbon dioxide emission is 660 million tons, which accounts for 4.7% of the total national emissions. In 2022, China’s Ministry of Industry and Information Technology (MIIT), Development and Reform Commission (DRC), and Ministry of Ecology and Environment (MOE) jointly issued the Implementation Plan for Carbon Peak Achievement in Non-ferrous Metals Industry (hereinafter referred to as the Implementation Plan). The goal of the Implementation Plan is to achieve carbon-peaking in the non-ferrous metal and other industries by 2030. Therefore, an in-depth analysis of the influencing factors of carbon emissions and an understanding of the decoupling status of carbon emissions in the non-ferrous metal industry are of great theoretical and practical significance for the early realization of the “double-carbon” target.

The natural resources in China are distributed unequally throughout the regions which has led to large differences in the size of the non-ferrous metal industry and carbon emissions among the provinces. Among them, Sichuan, Inner Mongolia, and Henan provinces have larger carbon emissions from non-ferrous metal extraction. The carbon emissions of three provinces, Shandong, Guangxi, and Henan account for 40% of the national carbon emissions from non-ferrous metal smelting and processing (Qu Qiushi, Wang & Xiang, 2021). In addition, the carbon dioxide emissions of the non-ferrous metal industry are mainly concentrated in the smelting of aluminum, copper, lead, zinc, and other metals (Zhang, 2021). Carbon emissions from the aluminum smelting industry account for about 80% of carbon emissions from the non-ferrous metal industry alone (Chaoxian, 2022).

There is a large amount of energy consumed during the smelting process, which is a major contributor to carbon emissions in the non-ferrous metals industry. Approximately 70% of China’s electricity sources are generated by burning fossil fuels such as coal, oil, and natural gas. Indirect carbon emissions from electricity consumption in aluminum and copper production account for 90% and 80% of total emissions, respectively (Gu Lin & Mingsheng, 2022). The aluminum smelting industry had an energy efficiency use of less than 50%, with the result being that the actual consumption of electricity in the aluminum metal production process is much higher than the theoretical value (Zheng Shili et al., 2022). Different processes of thermal copper smelting result in large differences in electricity consumption and carbon emissions (Liu et al, 2021). Inadequate energy utilization efficiency and outdated processes (Zeng et al., 2011) continue to increase energy consumption and energy intensity, further increasing carbon emissions. We sought to determine the decoupling status of scale growth and carbon emissions in the non-ferrous metal industry; the extent that factors such as energy structure and energy intensity affect carbon emissions in the non-ferrous metal industry; and the regional heterogeneity. These factors have great significance for the non-ferrous metal industry to realize low-carbon development and promote China’s carbon emission reduction cause.

Based on existing research, we evaluated the non-ferrous metal industry of 28 provinces (cities and autonomous regions) in China from 2000 to 2019. The panel interaction fixed effect model and the Tapio decoupling model were used to investigate the influencing factors of carbon emissions and the process of its decoupling status in the national and regional non-ferrous metal industries, respectively. The results show that: (1) China’s non-ferrous metal industry as a whole is in a negative decoupling state, and there is a large gap between the decoupling states of different regions; (2) energy structure, energy intensity, and cost have a positive impact on carbon emissions; (3) there is regional heterogeneity in the impact of non-ferrous metal production on the industry’s carbon emissions. We provide theoretical support for the implementation of policies to reduce carbon emissions in the non-ferrous metal industry in different regions.

The remainder of the article is organized as follows: ‘Literature Review’ provides a review of related studies and the contributions of this article. ‘Model Construction and Data Description’ describes the theoretical models and data sources used. ‘Empirical Analysis’ presents the main results of the study. ‘Conclusions and Policy Recommendations’ presents the conclusions and the related policy recommendations.

Literature Review

The reduction of carbon emissions is a global concern for scientists and progress has been made toward this goal. Economic growth (Yu et al., 2023), traditional energy consumption (Ma, Chen & Su, 2021), and energy intensity (Li et al., 2021) have been found to be direct factors leading to the deterioration of the natural environment and the increase of carbon emissions. Income inequality, on the other hand, contributes indirectly to the increase in carbon emissions by inhibiting the increase in carbon emission efficiency (Wang, Li & Li, 2023a). Green energy (Xu & Lin, 2023), technological innovation (Jianguo et al., 2022), economic globalization (Ali, Jianguo & Kirikkaleli, 2023), trade openness, and import diversification contribute to reducing carbon emissions (Wang, Wang & Li, 2023b; Wang, Zhang & Li, 2023c).

Research on the factors influencing carbon emission and the decoupling of high energy consumption and high emission industries has attracted the attention of many scholars. Jianguo, Cheng & Ali (2023) tested the carbon neutral hypothesis from the perspective of BRICS countries and found that financial innovation and economic growth promote carbon emissions in the transportation industry. Some scholars (Li, Li & Wang, 2022; Zhou, 2016) have used the decoupling index method to study the current status of carbon emissions and decoupling in China’s transportation sector. Their study found that the decoupling of China’s transportation carbon emissions and economic growth did not achieve a strong decoupling state, and was shown to be in a state of negative decoupling, declining decoupling, and expansionary negative decoupling in most years. The influencing factors of China’s industrial carbon emissions and their decoupling status were thoroughly studied using the LMDI decomposition method and the Tapio decoupling model (Yuan et al., 2022). The results show that energy consumption, energy intensity, capital stock, and labor input are the main factors influencing industrial carbon emissions and that carbon emissions and industrial economic scale are moving from a state of weak decoupling to strong decoupling. Based on the above study, it can be concluded that the main factors leading to the increase of carbon emissions are economic scale, energy intensity, and energy structure.

As an important part of the energy-intensive industries, the non-ferrous metal industry requires the consumption of large-scale energy during production and processing but also produces a large amount of carbon dioxide, which causes serious environmental pollution.

Based on the LMDI decomposition method and gray correlation analysis, the study by Shi Yuru (2014) found that economic scale, energy structure, and energy intensity (Chen et al., 2019) were found to have a significant contribution to carbon emissions in the non-ferrous metal industry, however, technological progress was shown to improve the environmental efficiency of the region (Chen & Lin, 2020). Zen (2011) used the Laspeyres index decomposition method to construct a decoupling measurement model of industrial development and carbon emissions. The results showed that the expansion of the industrial scale and the increase of the proportion of electricity consumption led to the increase of carbon emissions in the non-ferrous metal industry. The decoupling state has experienced four successive stages: strong recoupling, weak decoupling, expansive recoupling, and weak decoupling. Yang, Zhu & Huang (2018) used the Tapio model to study the decoupling state of economic growth and carbon emissions in China’s aluminum industry from 2006 to 2018. They found that the gap between the decoupling status of different regions is narrowing, and the decoupled regions are mainly found in the developed provinces in the central and eastern parts of the country.

The existing literature has provided an in-depth analysis of the factors influencing carbon emissions in the non-ferrous metal industry, which forms the basis for the subsequent research in this area. However, there are still some shortcomings: first, the research on the influencing factors of carbon emissions in the non-ferrous metal industry does not include the analysis between different periods and different regions. Second, the number of scholars involved in the research on the decoupling of carbon emissions in the non-ferrous metal industry is small, and the study period is often too far in the past. The non-ferrous metal industry has experienced rapid development in the last decade, and the results of past studies can no longer represent the decoupling status of the current non-ferrous metal industry. The marginal contributions of this study include: (1) the study period is divided into five development phases according to the layout of China’s Five-Year Development plan. The factors influencing carbon emissions of the country as a whole and of each region in different development phases were studied separately. The analysis of the regional heterogeneity of the factors that influence carbon emissions helps each region to formulate appropriate emission reduction policies according to its actual situation. (2) The empirical study on the decoupling of scale growth and carbon emission of the non-ferrous metal industry during the study period reveals the evolution of the decoupling status of carbon emission of the non-ferrous metal industry in China and each region over the past two decades.

Model Construction and Data Description

Tapio carbon decoupling model for the non-ferrous metal industry

The OECD and the Tapio decoupling models have been the most widely applied models among the studies on the decoupling of economic growth and environmental pollution (H, 2019; Zhang et al., 2022). In practice, the OECD decoupling model is highly sensitive to the base period, which may lead to errors in the process of determining the decoupling status (Zhou, 2016). In contrast, the Tapio decoupling model can effectively avoid the uncertainty generated by the selection of base period (Huan & Wang, 2019). It also has certain advantages in determining the succession process of the decoupling status. We chose the Tapio model to analyze the decoupling state of scale growth and carbon emissions and its succession process in the non-ferrous metal industry in China and its provinces. Based on the Tapio decoupling model (Tapio, 2005), the elasticity index of the decoupling of scale growth and carbon emission of the non-ferrous metal industry is calculated as follows: (1) ɛ=CO2,t−CO2,0CO2,0Productiont−Production0Production0=ΔCO2CO2ΔProductionProduction

where ɛ is the Tapio decoupling elasticity index of China and each province (municipality and autonomous region) in year t, which represents the decoupling relationship between non-ferrous metal production and carbon emissions; CO2,t , and CO2,0 represent the carbon emissions in year t and the base period, respectively; Productiont and Production0 represent the non-ferrous metal production in year t and the base period, respectively; ΔCO2 and ΔProduction represent the changes in carbon dioxide emissions and non-ferrous metal production during the study period, respectively. According to the Tapio decoupling model, the decoupling state of carbon dioxide emissions in the non-ferrous metal industry is divided into three basic states: decoupling, coupling, and negative decoupling, according to the ΔCO2, the ΔProduction and elasticity coefficient ɛ of the size of the decoupling state, the decoupling state is distinguished into eight different types of decoupling, as shown in Table 1. The decoupling status of China and each province in four time periods is shown in Table 2.

Table 1 Tapio decoupling type division.

Decoupling status	Decoupling type	ΔCo2	ΔProduction	Resilience index ɛ	Features	
Decoupling	Strong decoupling	<0	>0	ɛ<0	Non-ferrous metal production increases while carbon emissions decrease	
Weak decoupling	>0	>0	0<ɛ <0.8	Carbon emissions increase at a slower rate than production growth	
Recession decoupling	<0	<0	ɛ>1.2	Carbon emissions are decreasing faster than production	
Connections	Expansion of the connection	>0	>0	0.8<ɛ <1.2	Carbon emissions increase at the same rate as production growth	
Recession Connection	<0	<0	0.8<ɛ <1.2	Carbon emissions decrease at the same rate as production decreases	
Negative decoupling	Strong negative decoupling	>0	<0	ɛ<0	Increase in carbon emissions and decrease in production	
Weak negative decoupling	<0	<0	0<ɛ <0.8	Carbon emissions are decreasing at a slower rate than production	
Expansion negative decoupling	>0	>0	ɛ>1.2	Carbon emissions are increasing faster than the rate of increase in production	

Table 2 Decoupling state of non-ferrous metal production and carbon emissions in China and its 28 provinces (cities and autonomous regions), 2000–2019.

Region	2000–2004	2005–2009	2010–2014	2015–2019	
	ɛ	Decoupling Status	ɛ	Decoupling Status	ɛ	Decoupling Status	ɛ	Decoupling Status	
National	−0.405	Strong decoupling	0.729	Growth Connections	0.368	Growth Connections	1.984	Negative decoupling of growth	
Anhui	−0.995	Strong decoupling	−0.374	Strong decoupling	−0.307	Strong negative decoupling	−0.175	Strong decoupling	
Fujian	8.189	Negative decoupling of growth	−0.331	Strong decoupling	2.387	Negative decoupling of growth	0.523	Weak decoupling	
Gansu	−0.277	Strong decoupling	1.576	Negative decoupling of growth	0.312	Weak decoupling	0.411	Weak negative decoupling	
Guangdong	375.481	Negative decoupling of growth	1.383	Negative decoupling of growth	−2.568	Strong negative decoupling	2.759	Negative decoupling of growth	
Guangxi	39.034	Negative decoupling of growth	0.226	Weak decoupling	−75.954	Strong negative decoupling	0.106	Weak decoupling	
Guizhou	1.175	Growth Connections	0.011	Weak decoupling	0.732	Weak negative decoupling	−0.235	Strong decoupling	
Hebei	0.209	Weak decoupling	−0.757	Strong negative decoupling	0.254	Weak decoupling	−0.179	Strong negative decoupling	
Henan	0.240	Weak decoupling	1.825	Negative decoupling of growth	−5.975	Strong decoupling	−0.070	Strong negative decoupling	
Heilongjiang	−0.125	Strong negative decoupling	−0.984	Strong negative decoupling	0.326	Weak negative decoupling	0.548	Weak decoupling	
Hubei	3.835	Negative decoupling of growth	−0.079	Strong decoupling	−1.329	Strong decoupling	1.057	Weak negative decoupling	
Hunan	1.284	Negative decoupling of growth	0.427	Weak decoupling	−2.588	Strong decoupling	1.105	Recession Connection	
Jilin	0.518	Weak decoupling	−2.781	Strong negative decoupling	53.227	Negative decoupling of growth	−0.007	Strong decoupling	
Jiangsu	−11.230	Strong negative decoupling	0.194	Weak decoupling	−0.923	Strong negative decoupling	0.215	Weak decoupling	
Jiangxi	0.181	Weak decoupling	1.073	Growth Connections	0.351	Weak decoupling	16.196	Negative decoupling of growth	
Liaoning	0.987	Recession Connection	1.254	Negative decoupling of growth	0.764	Weak negative decoupling	−0.203	Strong decoupling	
Inner Mongolia	1.934	Strong decoupling	0.945	Growth Connections	−1.648	Strong decoupling	0.440	Weak decoupling	
Ningxia	−0.439	Strong decoupling	2.826	Negative decoupling of growth	−0.530	Strong decoupling	11.417	Recession decoupling	
Qinghai	−0.020	Strong decoupling	17.002	Negative decoupling of growth	0.129	Weak decoupling	−0.302	Strong decoupling	
Shandong	0.374	Weak decoupling	0.317	Weak decoupling	0.460	Weak decoupling	17.103	Negative decoupling of growth	
Shanxi	2.428	Negative decoupling of growth	−0.199	Strong decoupling	38.212	Negative decoupling of growth	69.651	Negative decoupling of growth	
Shaanxi	0.679	Weak decoupling	−0.356	Strong decoupling	−0.162	Strong decoupling	24.257	Negative decoupling of growth	
Shanghai	21.843	Negative decoupling of growth	−133.329	Strong negative decoupling	−0.113	Strong negative decoupling	0.279	Weak negative decoupling	
Sichuan	0.234	Weak decoupling	4.299	Negative decoupling of growth	−0.749	Strong negative decoupling	−0.562	Strong decoupling	
Tianjin	34.130	Recession decoupling	6.507	Negative decoupling of growth	0.662	Weak decoupling	0.212	Weak negative decoupling	
Xinjiang	−0.707	Strong decoupling	−1.215	Strong decoupling	−0.007	Strong decoupling	62.368	Negative decoupling of growth	
Yunnan	0.180	Weak decoupling	0.309	Weak decoupling	−0.091	Strong decoupling	0.282	Weak decoupling	
Zhejiang	0.625	Weak decoupling	1.029	Growth Connections	0.199	Weak negative decoupling	−0.202	Strong decoupling	
Chongqing	−0.681	Strong decoupling	0.800	Weak decoupling	0.727	Weak decoupling	7.430	Recession decoupling	

Panel interactive fixed effects model of factors influencing non-ferrous metal carbon emissions

The decoupling elasticity coefficient can only indicate the decoupling status of non-ferrous metal production growth and carbon emissions; it cannot analyze the driving factors of carbon emissions in the non-ferrous metal industry. Therefore, in order to further investigate the influencing factors of carbon emissions in the non-ferrous metal industry, we constructed a panel interaction fixed effects model of non-ferrous metal carbon emissions. The interaction fixed effects model controls the unobservable factors that change individually over time by setting the interaction term between individual and time effects, thus reducing the endogeneity problem of the model estimation, and improving the goodness of fit. Referring to the interaction fixed effects model (Eqs. (2) and (3)) of individual and time constructed by Bai (2009), the model is set as follows: (2) Yit=Xitβ+μit

(3) uit=λiFt+ɛiti=1,2,…,N,t=1,2,…,T

where Xit denotes the vector of explanatory variables, and β denotes a vector of coefficients to be estimated. μit denotes a factor structure, and λi represents a vector of factor loadings with different responses of regions to different factors. Ft represents a vector of factors that vary only over time. ɛit is the error term. λi , the Ft and ɛit are all unobserved.

Existing research results show that the industrial scale (Zen, 2011) and energy intensity (Chen et al., 2019; Shi Yuru, 2014) promote the increase of carbon emissions. However, improving the energy structure (Gu Lin & Mingsheng, 2022) and increasing the proportion of renewable energy use can help reduce carbon emissions. In addition, the technological innovation effect generated by increasing R&D costs (Chaoxian, 2022) encourages the development of low-carbon emissions technologies. The effects of energy structure, energy intensity, industrial scale, and costs on the carbon emissions of the non-ferrous metal industry in China and in each region, and the heterogeneity of the effects of these factors over time are described in Table 3. Before calculating the energy structure and energy intensity, all types of energy consumption are converted to standard coal consumption. To avoid the effects of heteroskedasticity in the model, all variables are logarithmized. The improved panel interaction fixed effects model is given in the following equation: (4) lnCit=β1lnESit+β2lnEIit+β3lnCostit+β4lnProductionit+μit

(5) uit=λiFt+ɛiti=1,2,…,N,t=1,2,…,T

where i denotes the province (city, autonomous region). i = 1, 2, …, 28; t denotes the time; Cit denotes the total carbon emissions of the non-ferrous metal industry in each region; ESit denotes the energy structure of the non-ferrous metal industry in each region, measured by the ratio of fixed energy consumption to total energy consumption; EIit denotes the energy intensity of the non-ferrous metal industry in each region, measured by the ratio of total energy consumption to total profit; Costit denotes the cost of the non-ferrous metal industry in each region, measured by the total operating cost of the non-ferrous metal industry; Productionit denotes the output of the non-ferrous metal industry in each region.

Table 3 Descriptive statistics of variables.

Variables	Variable Description	Unit	Average value	Standard deviation	Minimum value	Maximum value	
C it	Total carbon emissions	million tons	316.08	464.99	6.39	3302.05	
ES it	Energy mix	%	0.52	0.26	0.02	0.99	
EI it	Energy intensity	Million tons/billion yuan	9.79	166.42	−3,089.37	1,812.57	
Cost it	Cost	billion yuan	1,031.64	1,371.49	0.74	7,493.21	
Production it	Capacity	million tons	111.23	148.28	0.01	1,048.63	

Description of data

The data on carbon emission data and energy consumption the non-ferrous metal industry were taken from the China Carbon Accounting Database (CEADs). According to the statistical caliber of the CEADs, we only analyzed the carbon emissions of the non-ferrous metal industry in Scope 1 and the types of fossil energy consumption. Energy consumption in Scope 1 includes raw coal, cleaned coal, other washed coal, briquettes, coke, coke oven gas, other gas, other coking products, crude oil, gasoline, kerosene, diesel oil, and fuel oil. Data on total profit, cost, and non-ferrous metal output in the non-ferrous metal industry were obtained from the China Industrial Statistical Yearbook and provincial statistical yearbooks. Among them, the output variable of the non-ferrous metal industry uses ten commonly used non-ferrous metal output data instead, as these account for more than 99.5% of the total output of the non-ferrous metal industry. Our study focuses on the period between 2000–2019 in China in 28 provinces (municipalities and autonomous regions) due to the lack of data in Beijing, Hainan Province, and the Tibet Autonomous Region.

Empirical analysis

Trends in energy consumption, carbon emissions, and production changes in the non-ferrous metal industry

Due to the wide variety of energy sources used by the non-ferrous metal industry, we categorized the 17 main energy sources as solids, liquids, and gases. Solid energy included raw coal, cleaned coal, other washed coal, briquettes, coke, and other coking products, liquid energy includes crude oil, gasoline, kerosene, diesel oil, fuel oil, other petroleum products, and gas energy includes coke oven gas, other gas, other petroleum products. Liquid energy included crude oil, gasoline, kerosene, diesel oil, fuel oil, and other petroleum products, and gas energy included coke oven gas, other gas, LPG, refinery gas, and natural gas. The total energy consumption of the non-ferrous metal industry and the proportion of each type of energy used are shown in Figs. 1 and 2.

During the study period, energy consumption decreased in 2003 and 2011 (Fig. 1), however, the total energy consumption of all types in the non-ferrous metal industry was on an upward trend. The total energy consumption in 2000 was about 24 million tons of standard coal. In 2019, the total energy consumption reached about 83 million tons of standard coal, which was approximately 3.46 times the total energy consumption in 2000. As shown in Fig. 2, the proportion of solid energy consumption decreased year over year, from 78% in 2000 to 28% in 2019. In contrast, the share of gas energy consumption to all energy consumption increased year over year, from 8% in 2000 to 62% in 2019. Liquid energy consumption did not change much and was constant between 8% and 20%.

Figure 3 shows the changes in carbon emission and production in China’s non-ferrous metal industry from 2000 to 2019. Carbon emissions were higher 2000–2002 and remained at the level of 60 million tons. Carbon emissions then suddenly decreased in 2003, and the growth rate of carbon emissions was approximately −40%. This may have been due to an outbreak of SARS in November 2002, which led to the shutdown of production in most parts of the country, resulting in a decrease in carbon emissions from the non-ferrous metal industry. From 2003 to 2019, the annual growth rate of carbon emissions has been and remains positive. Carbon emissions in 2003 were 33 million tons and rose to 148.62 million tons in 2019, an increase of about 4.5 times. There has been a consistent trend in annual growth from 2000 to 2019 with an increase in output from 7.6 million tons in 2000 to 58.4 million tons in 2019. This reflects an increase of 7.68 times in the industry scale. The carbon emissions of the non-ferrous metal industry have maintained roughly the same growth rate as production over the study period.

Figure 1 Changes in energy consumption in China’s non-ferrous metals industry, 2000–2019.

Figure 2 Change in the proportion of various types of energy consumption in China’s non-ferrous metals industry, 2000–2019.

Figure 3 Change in carbon emissions and production in China’s non-ferrous metals industry, 2000–2019.

Analysis of the decoupling index between non-ferrous metal production growth and carbon emissions

Characteristics of the temporal evolution of the decoupled state of non-ferrous metal production growth and carbon emissions

We divided the time evolution characteristics of the decoupling state of carbon emissions in the non-ferrous metal industry into six categories for analysis, and the results are shown in Table 4. There are more regions with negative decoupling evolution characteristics, including Gansu and Hebei, and the decoupling state of carbon emissions in China’s non-ferrous metal industry has evolved from a positive decoupling state to a negative decoupling state. This is followed by regions characterized by decoupling-decoupling evolution, such as Anhui, Ningxia, and Qinghai. There are equal numbers of regions characterized by negative decoupling-decoupling and negative decoupling-negative trends, such as Fujian, Guangxi, and Guangdong. Guizhou and Liaoning experienced the evolution from a connected to a decoupled state, while Hunan experienced the change from a negative decoupled to a connected state. The results show that the temporal evolution characteristics of carbon decoupling in China’s non-ferrous metal industry are quite different among regions, and most of the regions are still in the negative decoupling state.

Table 4 Characteristics of carbon emission decoupling time evolution in the non-ferrous metals industry.

Evolutionary characteristics	Region	
Decoupling—decoupling	Anhui, Ningxia, Qinghai, Chongqing, Zhejiang, Jilin, Sichuan, Inner Mongolia	
Decoupling—Negative decoupling	Gansu, Hebei, Henan, Tianjin, Shandong, Shaanxi, Xinjiang, Yunnan, Jiangxi	
Negative decoupling—decoupling	Fujian, Guangxi, Heilongjiang, Jiangsu	
Negative decoupling—Negative decoupling	Guangdong, Shanghai, Hubei, Shanxi	
Connection—decoupling	Guizhou, Liaoning	
Negative decoupling—connection	Hunan	

Characteristics of the spatial evolution of the decoupled state of carbon emissions and industrial growth in the non-ferrous metal industry

For the convenience of analysis, China’s 28 provinces (municipalities and autonomous regions) are divided into eastern, central, and western regions. The eastern region includes Tianjin, Hebei, Liaoning, Shanghai, Zhejiang, Fujian, Shandong, Guangdong, and Jiangsu. The central region includes Shanxi, the Inner Mongolia Autonomous Region, Jilin, Heilongjiang, Anhui, Jiangxi, Henan, Hubei, and Hunan. The western region includes the Guangxi Zhuang Autonomous Region, Chongqing Municipality, Sichuan, Guizhou, Yunnan, Shaanxi, Gansu Province, Qinghai Province, Ningxia Hui Autonomous Region, and the Xinjiang Uygur Autonomous Region. As shown in Table 2, there are large differences in the decoupling status of carbon emissions from the non-ferrous metal industry in 28 provinces (municipalities and autonomous regions) in China.

In the first phase (2000–2004), provinces with strong decoupling of carbon emissions were mainly located in the western region. These locations included Gansu, Qinghai and Chongqing, and Anhui in the central region. Provinces with weak decoupling were evenly distributed in the eastern, central, and western areas, such as Zhejiang, Henan, and Yunnan. Provinces with negative growth decoupling and strong decoupling were mainly located in the eastern and central regions; these included the Fujian, Guangdong, and Hunan provinces. In addition, the carbon emissions in Tianjin, Liaoning, and Guizhou provinces were in recessionary decoupling, recessionary linkage, and growth linkage, respectively.

In the second stage (2005–2009), as the economy developed, the number of provinces with strong decoupling and weak decoupling of carbon emissions from the non-ferrous metal industry decreased. However, the Fujian and Hubei provinces changed from growth-negative decoupling to strong decoupling. There was an increase in the number of provinces with negative and strong decoupling growth, e.g., Gansu, Henan, Qinghai, and Sichuan. These have changed from decoupling to negative decoupling in the past. Jiangxi, Zhejiang, and the Inner Mongolia Autonomous Region have changed from a decoupling to growth-led status.

In the third stage (2010–2014), the decoupling status of carbon emissions from the non-ferrous metals industry in all provinces shifted to two broad types of decoupling and negative decoupling. Provinces such as Henan, Hebei, and Qinghai shifted from growth negative decoupling or strong negative decoupling in the previous stage to a strong or weak decoupling status. Provinces such as Anhui, Fujian, the Jiangsu Guangxi Zhuang Autonomous Region, and Guizhou moved from strong or weak decoupling in the previous period to strong negative decoupling, weak negative decoupling, or negative growth decoupling.

In the fourth phase (2015–2019), the decoupling status of carbon emissions from the non-ferrous metals industry in each region was still mainly divided into two broad types: decoupling and negative decoupling. Many provinces, such as Anhui, Guizhou, and Jilin, moved from negative decoupling in the previous phase to strong or weak decoupling. Qinghai, Yunnan, and the Inner Mongolia Autonomous Regions remained strongly or weakly decoupled. Several provinces, including Hebei, Henan, Shandong, and the Xinjiang Uygur Autonomous Region developed into a negative decoupling classification. However, the Shanghai Municipality, Shanxi Province, and Guangdong Province remained negatively decoupled. The Hunan Province also moved from a strong decoupling status in the previous period to a declining linkage status.

Our analysis shows that at the early stage of development (the first stage), most of the provinces with the decoupling of carbon emissions from the non-ferrous metal industry that were in a state of strong decoupling or weak decoupling were located in the central and western regions. These results are consistent with those of Zen (2011). However, provinces classified as negative decoupling were mostly located in the eastern and central regions. In the middle stages of development (Stage 2 and Stage 3), the decoupling status of carbon emissions in most provinces alternated between decoupling and negative decoupling. In the late stage of development (Stage 4), provinces with strong negative decoupling, weak negative decoupling, or negative growth decoupling were mainly located in the central and western regions.

Analysis of influencing factors of non-ferrous metal industry’s carbon emission

Analysis of carbon emission influencing factors of the non-ferrous metal industry at different stages

We found that the decoupling statuses varied during different stages of development and between different regions. The study period was divided into four stages: 2000–2004, 2005–2009, 2010–2014, and 2015–2019, and used the fixed effect model (Eqs. (4) and (5)) to conduct an in-depth study on the factors affecting the carbon emissions of the non-ferrous metal industry. The regression results are shown in Table 5.

Table 5 Regression results of carbon emission influencing factors of the non-ferrous metal industry by stages.

Variables	2000–2004	2005–2009	2010–2014	2015–2019	
	lnC it	lnC it	lnC it	lnC it	
lnES it	0.910***	0.882***	0.775***	−0.032	
	(0.240)	(0.096)	(0.108)	(0.075)	
lnEI it	0.520***	0.247***	0.400***	0.084*	
	(0.059)	(0.046)	(0.044)	(0.046)	
lnCost it	0.389***	0.371***	0.217*	0.099	
	(0.114)	(0.113)	(0.118)	(0.136)	
lnProduction it	0.835***	−0.033	−0.124*	0.049	
	(0.117)	(0.064)	(0.066)	(0.069)	
_cons	−0.855	1.306	4.652***	3.998***	
	(0.529)	(1.755)	(0.718)	(1.119)	
N	106	135	135	108	
F-statistic	349.47***	35.64***	90.55***	19.28***	
Notes.

***, **, * denote 1%, 5%, and 10% significance levels, respectively; Standard errors are in parentheses.

During the periods from 2000–2004, 2005–2009, and 2010–2014, energy structure (ES) had a significant positive impact on carbon emissions, and the impact coefficients decreased sequentially to 0.91, 0.882, and 0.775, respectively. From 2015 to 2019, the impact of the energy structure on carbon emissions was not significant. This may be due to the fact that the share of coal energy consumption decreased significantly during the study period, while the share of liquid energy and gas energy consumption increased significantly, resulting in a sequential decrease in the impact of the energy mix on carbon emissions.

Throughout the study period, energy intensity (EI) had a significant positive impact on carbon emissions, with impact coefficients of 0.52, 0.247, 0.4, and 0.084, respectively. The impact coefficients generally showed a gradually decreasing trend. This indicates that the impact on carbon emissions of the non-ferrous metal industry gradually decreases as energy intensity is reduced. However, as long as there is fossil energy consumption, the impact of energy intensity on carbon emissions still exists.

The cost (Cost) had a significant positive impact on carbon emissions during the periods of 2000–2004, 2005–2009, and 2010–2014, and the impact coefficient decreased in order as 0.389, 0.371, and 0.217, respectively. The impact of cost on carbon emissions was not significant from 2015 to 2019. However, this is because most of the costs in the non-ferrous metal industry are used for energy consumption in the production process (Chaoxian, 2022). This effect may also be due to the diminishing marginal utility of R&D costs invested in technological innovations, and the impact on carbon emissions in the non-ferrous metal industry.

From 2000 to 2004, the effect of production (Production) volume on carbon emission in the non-ferrous metal industry was significantly positive. This may be due to high carbon emissions from less technologically advanced machinery in the production process. From 2010 to 2014, the increase in production had a significant inhibitory effect on carbon emissions, probably due to the effect of technological progress and the elimination of backward production capacity.

Analysis of carbon emission influencing factors of non-ferrous metal industry in different regions

In order to study the regional differences in the carbon emissions of non-ferrous metals due to different drivers, we analyzed the spatial heterogeneity of the influencing factors of carbon emissions of non-ferrous metals at the national level, in the east, in the center, and in the west, respectively. The results of the analysis are shown in Table 6.

At the national level, energy structure and intensity, cost, and production all had a significant positive effect on carbon emissions, with coefficients of 0.454, 0.413, 0.238, and 0.073, respectively. This is consistent with the findings of Chen et al. (2019) and Shi Yuru (2014), who concluded that the expansion of the industrial scale is the main factor contributing to the increase in carbon emissions. Improvements in the energy structure and an increased use of renewable energy can help to reduce carbon emissions by reducing energy intensity (Zhang, 2021).

The three major regions showed the following differences: (1) The energy structure of the eastern and central regions made significant contributions to the carbon emissions of the non-ferrous metal industry, and the coefficient of the energy structure of the western region was larger than that of the eastern region (0.720>0.198). This may be because the proportion of solid energy consumption in the eastern region was smaller than that in the western region. On the other hand, the differences that lead to carbon emissions in the non-ferrous metal industry are closely related to the energy conditions in each region (Wu, Gao & Zhang , 2022). (2) Energy intensity and cost had a significant positive effect on the carbon emissions of the non-ferrous metal industry in the eastern, central, and western regions. (3) In the eastern region, the output of the non-ferrous metal industry had a significant inhibitory effect on carbon emissions, with a coefficient of −0.228. This may be because economically developed regions in the face political and economic pressures due to carbon emissions, and the local governments restrict the production of non-ferrous metals to reduce the increase of carbon emissions in the future. For example, the plan on the transformation and upgrading of the non-ferrous metal industry issued by Zhejiang Province mentions that it will encourage the transfer of ordinary copper processing projects outside the province and abroad, while the province will develop the sectors of deep processing, research and development, and marketing sectors.

Table 6 Regression results of carbon emission influencing factors of the non-ferrous metals industry by region.

Variables	National	Eastern
Region	Central
Region	Western
Region	
	lnC it	lnC it	lnC it	lnC it	
lnES it	0.454***	0.198***	0.181	0.720***	
	(0.051)	(0.069)	(0.111)	(0.067)	
lnEI it	0.413***	0.665***	0.422***	0.288***	
	(0.025)	(0.033)	(0.046)	(0.041)	
lnCost it	0.238***	0.811***	0.188***	0.295***	
	(0.051)	(.047)	(0.071)	(0.084)	
lnProduction it	0.073**	−0.228***	0.046	0.124	
	(0.033)	(0.055)	(0.052)	(0.080)	
_cons	−8.167**	0.393	8.094**	3.113***	
	(3.201)	(0.294)	(3.466)	(0.526)	
N	484	177	160	147	
F-statistic	79.21***	503.74***	29.94***	405.71***	
Notes.

***, **, * denote 1%, 5%, and 10% significance levels, respectively; Standard errors are in parentheses.

Conclusions and Policy Recommendations

Conclusion

At present, the global carbon dioxide emission problem is becoming increasingly more serious. There have been many studies on the influencing factors of carbon emission of high energy-consuming and high-emission industries. However, there are few empirical studies on the carbon emission influencing factors of the non-ferrous metal industry, especially the analysis of the decoupling state between the scale growth and carbon emission of the non-ferrous metal industry. Therefore, we measured the decoupling status of the carbon emissions of the non-ferrous metal industry in China and each region based on the data related to the non-ferrous metal industry in 28 provinces from 2000 to 2019 using the Tapio decoupling model. In addition, this study analyzed how energy outcomes, energy intensity, industry scale, and cost affect carbon emissions in the non-ferrous metal industry using a panel interaction fixed effects model. The main findings were as follows:

1. Although the proportion of solid energy such as coal consumed by China’s non-ferrous metal industry is decreasing, the total amount of fossil energy consumed is still increasing. The scale of the non-ferrous metal industry and carbon dioxide emissions continue the trend of increasing year by year.

2. During the period under study, the decoupling status of carbon emissions in China’s non-ferrous metal industry experienced three stages of strong decoupling-growth. In addition, the growth rate of carbon emissions in the non-ferrous metal industry was higher than the growth rate of production in the fourth development period. From the perspective of different regions, the decoupling status of carbon emissions in some provinces in the eastern and central regions changed from negative decoupling to strong or weak decoupling, and the decoupling status is improving. However, the decoupling status of carbon emissions in most provinces in the western region is deteriorating from the initially strong decoupling or weak decoupling to strong negative decoupling or growth negative decoupling status. The results show that there are large differences in the decoupling status of carbon emissions in the non-ferrous metal industry between different regions.

3. Among the influencing factors of carbon emissions in the non-ferrous metal industry, energy structure, energy intensity, and cost have a significant positive effect on carbon emissions in all periods. As far as different regions are concerned, energy intensity and cost have a significant contribution to carbon emissions in the non-ferrous metal industry in the eastern, central, and western regions. The energy structure makes a positive contribution to carbon emissions in the eastern and western regions. In addition, the output of the non-ferrous metal industry has an uncertain effect on carbon emissions. Specifically, production has a positive effect on carbon emissions in the early stage and a negative effect in the later stage. In terms of different regions, production has a significant negative effect on carbon emissions only in the eastern region. The results show that there are large regional differences in the influencing factors of carbon emissions in the non-ferrous metal industry.

Policy recommendations

Based on the above findings, this study makes the following policy recommendations. (1) The carbon emissions of the non-ferrous metal industry are mainly caused by the direct or indirect consumption of large amounts of fossil energy and electricity in the smelting process, while China’s power generation structure is still dominated by thermal power. Therefore, the non-ferrous metal industry should reduce the consumption of fossil energy, increase the proportion of renewable energy, optimize the energy structure, and achieve the reduction of carbon emissions from the source. However, local governments should make full use of local resource endowment conditions, and the western and eastern coastal areas should continue to expand the scale of photovoltaic, wind power, and other clean energy generation. In addition, the construction of inter-regional ultra-high-voltage transmission lines should be encouraged. This will provide clean energy from the western regions to the central and eastern regions, encourage the non-ferrous metal industry to get rid of the status quo of consuming fossil energy, and promote the development of low-carbon energy resources.

(2) The decoupling status of the scaled growth of the non-ferrous metal industry and carbon emissions varies widely among regions, but most provinces are still in a negative decoupling status. Therefore, the local governments should strengthen the resolute elimination of backward production capacity, and guide the transformation of high-polluting, energy-consuming old technology, and uncompetitive non-ferrous metal enterprises to switch to production or exit. The governments should promote the acceleration of upgraded industrial structures in developing non-ferrous metal products and finishing development. In addition, increasing investment in research and development of low-carbon technology in the non-ferrous metal industry is equally important. The regeneration of waste aluminum, short-range lithium materials, and other low-carbon technology can achieve the recycling of non-ferrous metals and greatly reduce carbon emissions in the non-ferrous metal smelting link. (3) Energy intensity and cost are also important factors affecting carbon emissions in the non-ferrous metal industry. We should accelerate the research and development and popularization of energy-efficient production processes to realize the purpose of saving resources and reducing energy intensity. For example, large-scale pre-baked electrolytic cell technology is preferred in the production of electrolytic aluminum. There also needs to be a focus on improving the environmental management systems and carbon emissions trading mechanism. By increasing the cost of pollution and carbon emissions, enterprises will be forced to carry out green technology innovations and ultimately realize the technological upgrading of the non-ferrous metal industry and low-carbon development.

Research gaps and future recommendations

This study has furthered the research on carbon emissions in the non-ferrous metal industry, however, there are additional recommendations for future research. The focus of this research was on the carbon emissions of China’s non-ferrous metal industry and its decoupling, and it does not consider the specific situation of other countries or regions. Therefore, the carbon emissions of other large non-ferrous metal industry countries should be studied in the future. The carbon emission situation of the non-ferrous metal industry was only considered in specific provinces, while the carbon emissions of neighboring provinces may also have an effect. Therefore, the spatial spillover effect of carbon emissions from the non-ferrous metal industry and its influencing factors should also be examined in the future. Finally, under the background of the “double carbon” target, the research on carbon emission forecasting of the non-ferrous metal industry is also of practical significance for the early realization of the low-carbon development of the non-ferrous metal industry.

Supplemental Information

Supplemental Information 1 Original data

Click here for additional data file.

Additional Information and Declarations

Competing Interests

Author Contributions

Data Availability

All authors consider that there are no competing interests.

Guohua Zeng conceived and designed the experiments, performed the experiments, authored or reviewed drafts of the article, and approved the final draft.

Minglong Zhong conceived and designed the experiments, performed the experiments, analyzed the data, prepared figures and/or tables, authored or reviewed drafts of the article, and approved the final draft.

Chengzhang Xiao analyzed the data, authored or reviewed drafts of the article, and approved the final draft.

The following information was supplied regarding data availability:

The original data is available in the Supplementary File.

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
