# Peer review of "Drivers and decoupling analysis of carbon emissions in the non-ferrous metal industry-evidence from 28 provinces in China"

_PeerJ, doi:10.7717/peerj.16575_

## Round 0.1 · original submission · Major Revisions

The paper has merits but needs extensive revisions. Please incorporate all concerns of the reviewers and please submit the revised version along with a point-to-point rebuttal letter. In addition, the introduction section should include the literature gap and the contribution of the study in a clear manner. The reviewers have suggested some citations. Please consider these studies if relevant to the study. Otherwise, you may ignore the suggested citations.

**Language Note:** The review process has identified that the English language must be improved. PeerJ can provide language editing services - please contact us at [email protected] for pricing (be sure to provide your manuscript number and title). Alternatively, you should make your own arrangements to improve the language quality and provide details in your response letter. – PeerJ Staff

·

Basic reporting

Thank you for submitting your paper for consideration. After carefully reading the paper titled “Analysis of drivers and decoupling of carbon emissions in the non-ferrous metals industry-evidence from 28 provinces in China”, The following issues need to be addressed before publication:
1. The paper's abstract must be improved according to why, how, and for whom this study was conducted. The quality of the abstract is not up to the mark and needs improvement to reach scientific writing merit. It must be high standard of writing to publish in outstanding journals like this.
2. The authors should reorganize the structure of the introduction section to thoroughly express the aspects of this study including the background, current progress, motivation, the research question, the objective, the contribution, etc.
3. Motivation and the contribution need to be strengthened. The contribution of this study needs more logical detail. The contribution of this paper is not well explained in the current version of the paper.
4. The authors should take into account the most recent information regarding the focal variable in the literature. To enhance the discussion in the literature, please include recent studies 2021-2023. I suggest that you expand your review by citing the most recent study,
i. https://doi.org/10.1007/s11356-022-19763-1
ii. https://doi.org/10.1016/j.egyr.2023.03.040
iii. https://doi.org/10.1002/gj.4732
5. All equations need appropriate references in the text.
6. The conclusion section can be improved. Policy implications are not sufficiently discussed. It is also suggested to provide some of the limitations pertaining to the current study and future directions for research.
7. Modify the paper based on the journal’s guidelines, especially the references.
8. Please check the manuscript again for errors.

Experimental design

no

Validity of the findings

no

Additional comments

no

Reviewer 2 ·

Basic reporting

It is my pleasure to review the manuscript for the PeerJ, an esteemed and uprising journal. In the manuscript of “Analysis of drivers and decoupling of carbon emissions in the non-ferrous metals industry - evidence from 28 provinces in China”, the authors investigated spatial and temporal evolution trends of the decoupling status of carbon emissions in China's non-ferrous metal industry in each province using the Tapio decoupling model, and then examined the factors inûuencing carbon emissions in China's non-ferrous metal industry using panel data based on data of China's non-ferrous metal industry from 2000-2019. The work presented is relevant to the Journal's field. The manuscript has got some potential. I would like to congratulate the authors for a considerable amount of work that they have done. Especially, the authors reported that both environmental information disclosure and green technology innovation have a positive effect on carbon efficiency, and their impact is complementary. In addition, the authors also uncovered the equity heterogeneity, with environmental information disclosure having a more significant impact on state-owned enterprises, and green technology innovation having a more significant impact on non-state-owned enterprises. This manuscript has provided a new case to better understanding urban-rural hierarchies with spatial quantile regressions. However, the manuscript needs further improved before to be accepted for publication. The reviewer has listed some specific comments that might be helpful of the authors to further enhance the quality of the manuscript. Please consider the particular comments listed below:

1, The authors need to improve the abstract. Therefore, the abstract should answer these questions about your manuscript: What was done? Why did you do it? What did you find? Why are these findings useful and important? Answering these questions lets readers know the most important points about your study and helps them decide whether they want to read the rest of the paper. Make sure you follow the proper journal manuscript formatting guidelines when preparing your abstract.

2, sections of Introduction, literature reviews. Although these two sections are well-structured and well-organized, the novelty of this paper should be further justified by highlighting main contributions to the existing introduction and literature review. This could be clearly presented in your related work. Please consider citing following papers entitled “Does urbanization redefine the environmental Kuznets curve? An empirical analysis of 134 Countries”; and entitled “Free trade and carbon emission revisited: The asymmetric impacts of trade diversification and trade openness”; and entitled “Trade protectionism jeopardizes carbon neutrality–Decoupling and breakpoints roles of trade openness”, and entitled “The impact of energy efficiency on carbon emissions: Evidence from the transportation sector in Chinese 30 provinces”; and entitled “Per-capita carbon emissions in 147 countries: The effect of economic, energy, social, and trade structural changes”; and entitled “Uncovering the impact of income inequality and population aging on carbon emission efficiency: An empirical analysis of 139 countries”. There has already been a large number of literatures related to your research topic, i.e. carbon emission. Therefore, it should be better elaborate the contribution of the work to the existing literature, so as to further bridge the gaps between the research background and research purposes.

3. The methods could benefit from further improvement. It is important to provide a clear justification for the methodology approach used, explaining why it was chosen and how it is
appropriate for the research question at hand. Additionally, it would be helpful to reference prior studies that have successfully used this methodology approach to strengthen the
argument for its use in this particular study.
- The data section requires improvement. The authors must address several key questions to provide a better understanding of their approach. Specifically, why were these variables selected for the model? What does the existing literature say about these variables? Additionally, it's important to provide information on previous authors who have used these variables. Without this information, readers may find it difficult to fully comprehend the approach and results
presented in the study.

4. section of Results. The section is well-structured and well-organized. However, it would be better to discuss what your findings are different from the past works. A comparison with the results of the previous paper would further enhance the innovative nature of the paper.

5- The authors need to improve the quality of the conclusions section. The conclusions section needs to be supported by the results and the authors need to show how their investigation advances the field from the present state of knowledge.
- To provide more comprehensive and actionable recommendations, the authors should create a dedicated subsection titled "Policy Implications." In this section, they should identify the specific areas where the current policy falls short and explain why their proposed recommendations can help improve the status quo. It's important to keep in mind that policymakers are interested in practical, cost-effective, and socially acceptable solutions, so the authors should address the
following questions before presenting their recommendations:
What specific changes need to be made?
How will these changes be implemented?
What resources will be required to implement the changes, and where will they come from?
What are the overall benefits of the proposed changes for policymakers and society as a whole? By answering these questions, the authors can provide a more compelling and practical set of policy recommendations that can help address the shortcomings of the current policy and lead to positive changes.
- The authors should consider creating a new subsection titled "Limitations and Future
Recommendations". It's essential to address the study's limitations, which are the design or
methodology constraints that may have affected the interpretation of the research findings.
Limitations may have an impact on the ability to generalize results or describe applications for practice, as well as the usefulness of the findings that resulted from the research design or
method used to establish internal and external validity, or unanticipated challenges encountered during the study.
In addition to addressing limitations, future recommendations should consider the following
aspects: (1) building upon a specific finding in the research; (2) addressing a flaw in the research design; (3) testing a theory, framework, or model in a new context, location, or culture; (4) re-
evaluating or (5) expanding a theory, framework, or model. It's important to consider these aspects to ensure that future research is based on solid foundations and provides valuable insights that can inform practice and policy decisions.
- The authors should prioritize improving the presentation quality of the manuscript,

6 particularly with regards to the organization of the text and the presentation of tables and
figures. While the manuscript shows promise, its overall investigation quality requires
improvement. The authors must ensure that the manuscript is both attractive and readable, in order to increase its likelihood of being read and cited. Paying close attention to details in all manuscripts will be critical to achieving this goal.

7, There are still some occasional grammar errors through the revised manuscript especially the article ''the'', ''a'' and ''an'' is missing in many places, please make a spellchecking in addition to these minor issues. In addition, some sentences are too long to be easy to read. It is recommended to change to short sentences, which are easier to read.

8, References. Please check the references in the text and the list; You should update the reference. Please read the latest published papers carefully and format your references according to the format required by PeerJ. If this revised paper is sent to me for re-review, the first thing I will check the references.

Experimental design

please see Basic reporting

Validity of the findings

please see Basic reporting

Additional comments

please see Basic reporting

Reviewer 3 ·

Basic reporting

Abstract: Too much long sentences. Point ii (line 48-52) of results in Abstract is confusing and very long sentence. Make it clear please. Policy implications should be from results obtained and not from imagination.
Introduction: it is very short. You needed to address, why you selected this topic to study? why it is important to conduct this study? What are the problems? What will happen if such studies will not be .
conducted? Write a complete statement of problem in the last paragraph of introduction.
Literature Review: Typo error in line 72. too much long paragraphs with many ideas. try to discuss one idea in one paragraphs and all paragraphs should be interconnected. Line 95: only one sentence paragraph. Only one good thing in Literature Review is that methodological literature discussed. You needed to address empirical and theoretical issues. Based on these issue derive statement of problem.

Experimental design

Accept my appreciation for writing good methodology. You described data well.

Validity of the findings

Results obtained as in methods and material discussed however these results are not justified from past literature.
Conclusion: It should not be in numbers but a brief summary of your findings. Combine both conclusion and policy recommendation. Policy implications should be from results obtained and not from imagination. Then, put these in the last lines of introduction.

---

## Round 0.2 · accepted · Accept

The paper has been improved incorporating all comments of reviewers and is accepted for publication.

·

Basic reporting

accepted for publication

Experimental design

no

Validity of the findings

no

Additional comments

no